# Astragalus Extract Mixture HT042 Alleviates Dexamethasone-Induced Bone Growth Retardation in Rat Metatarsal Bones

**DOI:** 10.3390/nu16142333

**Published:** 2024-07-19

**Authors:** Chae Yun Baek, JunI Lee, Donghun Lee, Hocheol Kim

**Affiliations:** 1Department of Herbal Pharmacology, College of Korean Medicine, Gachon University, 1342 Seongnamdae-ro, Sujeong-gu, Seongnam-si 13120, Republic of Korea; cyning20@gachon.ac.kr (C.Y.B.); lgl5278@gachon.ac.kr (J.L.); 2Department of Herbal Pharmacology, College of Korean Medicine, Kyung Hee University, 26 Kyungheedae-ro, Dongdaemun-gu, Seoul 02447, Republic of Korea

**Keywords:** HT042, dexamethasone, growth retardation, proliferation, hypertrophy, apoptosis

## Abstract

The most widely used synthetic glucocorticoid, dexamethasone (DEX), causes stunted growth in children when used excessively or for long periods of time; however, there are still plenty of pediatric patients require long-term treatment with DEX. As an alternative, growth hormone is used in combination, but it has side effects, a high cost, and psychological factors, and it is not satisfactory in terms of effectiveness. It is necessary to develop a safe and affordable treatment that can replace it. The Korean Food and Drug Administration approved HT042, a standardized functional food ingredient, with the claim that it can help height growth of children. In this study, it was found that HT042 activated the Indian hedgehog/parathyroid hormone-related protein signaling pathway and enhanced the number of growth hormone receptors and insulin-like growth factor-1 receptors on the growth plate surface, which were reduced by DEX treatment, and restored growth retardation. In metatarsal bone and primary chondrocyte models, it was found that HT042 can promote the length of growth plate and recover DEX-induced growth retardation. It was also found that HT042 promotes cell proliferation using bromodeoxyuridine and terminal deoxynucleotidyl transferase dUTP nick end labeling assays; moreover, we verified increased expression of GHR/IGF-1R and Ihh/PTHrP pathway activity using qRT-PCR, western blotting, and siRNA analyses to verify its direct action on the growth plate. The anti-apoptotic effect of HT042 was identified by regulating the expression of apoptotic factors such as caspase-3, Bcl2, Bclx, and Bax. These results were identified using both ex vivo and in vitro models. Our study verified that co-administration of HT042 could recover the DEX induced growth retardation

## 1. Introduction

The most widely used synthetic glucocorticoid, dexamethasone (DEX), is used to treat a variety of common pediatric conditions, such as asthma, psoriasis, and atopic dermatitis [1,2]. Among pediatric patients aged 18 years and younger, 43% have been prescribed DEX in the past year, and the incidence of adverse events including hyperparathyroidism, sex hormone deficiency, vitamin D deficiency, and ligament or muscle atrophy was 2.9 times higher compared with patients who were not treated DEX [3,4,5,6]. Excessive or prolonged use of DEX can lead to stunted growth as directly acts on the growth plate by downregulating the binding capacity of growth hormone (GH) and insulin-like growth factor-1 (IGF-1), reducing number of GH receptors (GHR) and IGF-1 receptors (IGF-1R) on the surface of the growth plate, and inhibiting the Indian hedgehog (Ihh)/parathyroid hormone-related protein (PTHrP) signaling pathway leading to decrease longitudinal bone growth in growing children [7,8,9]. Despite this, approximately 50% of patients in the pediatric population are inevitably treated with long-term DEX [10,11].

In the previous report, studies have attempted to treat growth retardation by administering GH to overcome growth retardation caused by DEX treatment [12,13]. However, possible side effects of combined DEX and GH treatment include insulin resistance, carbohydrate intolerance, increased incidence of diabetes, hyperglycemia, decreased renal function due to increased glomerular filtration rate, and extremity hypertrophy [14,15]. GH treatment is chosen as the first treatment option of DEX induced growth retardation, but due to side effects mentioned above and other factors such as high medical cost and psychological burden induced by daily injection, only 29.1% of patients are satisfied [4,16,17]. Therefore, there is a need to develop a safe, affordable, and convenient treatment that can prevent stunted growth induced by DEX.

Astragalus extract mixture HT042 is a standardized health functional food ingredient permitted by the Korean Food and Drug Administration (KFDA) for promoting children’s height growth. The basic mechanism of HT042-induced bone growth is through the enhancement of GH secretion, which leads to IGF-1 synthesis both systemically and locally [18,19]. HT042 stimulates chondrocyte proliferation in the growth plate and subsequent hypertrophy of the growth plate, leading to longitudinal bone growth [20]. It is expected that HT042 acts as a secretagogue for growth-acting paracrine factors, increasing the number of receptors and promoting the secretion of PTHrP, IGF, and several growth factors that are inhibited by DEX [17,21,22]. In this study, it was aimed to determine whether HT042 activates the Ihh/PTHrP signaling pathway to increase the number of GHR and IGF-1R on the growth plate surface and reverse the growth retardation caused by DEX treatment.

Because DEX not only systemically causes growth retardation but also inhibits growth by directly reducing the number of GHR/IGF-1Rs in the growth plate, it was chosen to use the metatarsal bone (MTB) model, which excludes systemic effects, instead of animal models that may have systemic effects [23,24,25]. The ex vivo organ culture model of fetal rat MTB is a physiologically well-known model that has been extensively used to study the endochondral ossification process during longitudinal bone development and the effects of various types of factors on postnatal growth [25,26,27]. The MTB allows for the characterization of growth responses ex vivo without being affected by systemic factors that may be received from the whole body [28]. Therefore, studies in the MTB are well suited to identify newly reported targets where HT042’s growth-promoting effects may directly affect chondrocyte number of GHR/IGF-1-R, proliferation, differentiation activity, and anti-apoptosis in addition to the canonical GH-IGF-1 axis identified in previous studies. In the present experiments, it was aim to determine whether direct DEX treatment of MTB restores growth retardation by increasing GHR and IGF-1R numbers and inducing the Ihh/PTHrP signaling pathway.

In this study, it was aimed to determine whether HT042 can counteract the ability of DEX to inhibit growth in a model in which MTB is treated with DEX to cause growth inhibition. The MTB was cultured to determine length growth and to evaluate the stimulatory effect of HT042 on chondrocyte proliferation. Proliferation was assessed using bromodeoxyuridine (BrdU) labeling after HT042 administration. In addition, it was measured mRNA and protein, and used small interfering RNA expression to determine if it reduces the number of GHR/IGF-1R and Ihh/PTHrP signaling pathway activity to determine if it restores proliferation and differentiation. In addition, to evaluate the effect of anti-apoptosis, terminal deoxynucleotidyl transferase dUTP nick end labeling and a BrdU assay were performed after HT042 administration.

## 2. Materials and Methods

### 2.1. High-Performance Liquid Chromatography (HPLC) Analysis

NeuMed Company (Seoul, Republic of Korea) provided the powder of HT042. The production procedure and quality monitoring for HT042 were performed in accordance with the guidelines provided by the KFDA. It was performed a chromatographic analysis on each sample using a Waters Company system (Waters corp, Milford, MA, USA). A reverse-phase SunFire C18 column (4.6 × 250 mm, 5 µm, Waters) was utilized. Standardized HT042 composed of 0.36% eleutheroside E, 0.15% shanzhiside methyl ester, and 0.008% formononetin. The analysis method was indicated in Table 1 and Table 2.

### 2.2. Animals

Female Sprague Dawley (SD) rat embryos at 20 days after conception were supplied by Samtako (Osan-si, Republic of Korea). The Gachon University Center of Animal Care and Use approved all of the experiments provided above (GU1-2022-IA0052-00).

### 2.3. Whole MTB Culture

At 20 days after conception, the 2nd, 3rd, and 4th MTB rudiments were obtained from SD rat fetuses and grown separately in 24-well plates. Every well included 0.5 mL of minimum essential medium (Gibco™, Norristown, PA, USA) with 0.05 mg/mL ascorbic acid (Sigma-Aldrich Co., Saint Louis, MO, USA), 1 mM sodium glycerophosphate (Sigma-Aldrich, USA), 0.2% bovine serum albumin (Sigma-Aldrich, USA) and 5% penicillin-streptomycin (Gibco, USA). MTB rudiments were cultured for 21 days in a 5% CO_2_ incubator on 37 °C. The medium was changed on day 2–3. During the 21-day culture period, MTB was cultured with HT042 and DEX.

### 2.4. Measurement of MTB Longitudinal Growth

The length of each MTB was measured using a dissection microscope with an eyepiece micrometer (1 mm stage micrometer). To evaluate the MTB growth rate, MTBs were recorded during the 21-day culture period with a dissecting microscope. For each treatment group, 90 MTBs separated from 15 rat embryos were used.

### 2.5. Growth Plate Height

MTB was preserved in 4% phosphate-buffered paraformaldehyde (PFA) for an overnight duration upon the end of the culture period. Next, 5 μm thick sections of PFA-fixed and paraffin-embedded MTB were prepared and deparaffined in xylene. Cresyl violet (CV; Sigma-Aldrich) and toluidine blue (TB; Sigma-Aldrich) stained chondrocytes. The overall growth plate height and zonal height were measured with ImageJ software 1.54d (NIH, New York, NY, USA) at two sections. From each of the two sections, it was measured the length of the proliferative zone (PZ) and hypertrophic zone (HZ) and calculated their values. The PZ in the MTB growth plate contained flattened cells distributed in columns parallel to the bone’s longitudinal axis. Large cells developed a layer in the HZ next to the major ossification center, which is the calcified portion of the MTB.

### 2.6. Immunohistochemistry (IHC) Analysis

MTBs were preserved in 4% PFA for an overnight duration upon the end of the culture period. A 5 μm thick section of MTB was PFA-fixed and paraffin-embedded. IHC was performed according to the guidelines of the Dako REAL^TM^ EnVision Kit (Agilent, Santa Clara, CA, USA), which will be described briefly. After 24 h, the coverslip was blocked with Dako REAL^TM^ peroxidase blocking solution (Agilent, USA) for 5 min and washed with wash buffer. The primary antibody, Ihh (Cell Signaling Technology, Danvers, MA, USA), was incubated overnight at 4 °C. The next day, after washing with wash buffer, it was reacted the sample with Dako REAL^TM^ EnVision^TM^/HRP (Agilent, USA) for 30 min at room temperature (RT). Then, it waswashed twice with wash buffer. It was reacted the sample with substrate working solution (Dako REAL^TM^ Substrate Buffer + Dako REAL^TM^ 3,3′-diaminobenzidine (DAB) + chromogen) and washed it with distilled water (DW). Lastly, it was mounted with Dako mounting medium (Agilent, USA).

### 2.7. Terminal Deoxynucleotidyltransferase-Mediated dUTP End Labeling (TUNEL) Assay

The in situ apoptosis detection kit was used to perform the TUNEL assay in accordance with the manufacturer’s instructions (Takara Biomedicals, Shiga, Japan). In short, the sections were permitted to incubate at room temperature for 15 min with 15 μg/mL proteinase K and then were rinsed with PBS. It was used 3% H_2_O_2_ to inactivate endogenous peroxidase for 5 min at RT. The sections were treated with a permeabilization buffer for 5 min at 4 °C in order to make them permeable. In a humid chamber at 37 °C for 90 min, several fragmented 3′-OH ends in the sections were labeled with digoxigenin-dUTP in the presence of terminal deoxynucleotidyltransferase and subsequently washed with phosphate-buffered saline. Afterwards, the sections were exposed to peroxidase-conjugated anti-digoxigenin antibody for 30 min at RT. The peroxidase–DAB reaction was used to visualize apoptotic nuclei.

### 2.8. Chondrocyte Culture

MTBs were separated from SD rat embryos at 20 days after conception, rinsed in PBS, and treated with 0.2% trypsin (Gibco, USA) for 1 h and 0.2% collagenase (Sigma-Aldrich, USA) for 4 h. The cell supernatant was removed, filtered using the 70 μm cell strainer (Corning Inc., New York, NY, USA), rinsed with warm PBS and then added to serum-free Dulbecco modified eagle medium (DMEM), and counted. Chondrocytes were cultured in 100 mm dishes at a density of 5 × 10^4^/cm^2^ in DMEM with penicillin-streptomycin (Gibco, USA), 50 μg/mL ascorbic acid, and 10% fetal bovine serum (FBS). The medium was replaced every 2–3 days.

### 2.9. Bromodeoxyuridine (BrdU) In Situ Incorporation Using Immunofluorescent (IF) Staining

Chondrocytes were incubated in a 96-well plate at a density of 5 × 10^3^/well in cultured media with 30 μg/mL HT042 and/or 100 ng/mL DEX. The IF was performed according to the guidelines of the Dako REAL^TM^ EnVision Kit (Agilent, USA), which will be described briefly. After 24 h, the coverslip was blocked with Dako REAL^TM^ peroxidase-blocking solution (Agilent, USA) for 5 min and washed with wash buffer. The primary antibody, BrdU (SantaCruz Inc., Santa Cruz, CA, USA), was incubated overnight at 4 °C. The next day, after washing with wash buffer, it was reacted it with Alexa Fluor 488-conjugated AffiniPure Goat Anti-Mouse IgG (H+L) (Assay Genie Corp., Seoul, Republic of Korea) for 30 min at RT. Then, it was washed twice with wash buffer. It was reacted Fluoroshield^TM^ with 4′,6-diamidino-2-phenylindole (DAPI) staining solution (Agilent, USA) for 10 min and washed with DW. Lastly, it was performed mounting with Dako mounting medium (Agilent, USA).

### 2.10. In Situ Cell Death

Chondrocytes were exposed with 30 μg/mL HT042 and/or 100 ng/mL DEX for 24 h. Apoptotic cells were detected using the in situ apoptosis detection kit (Takara, Japan) according to the manufacturer’s instructions.

### 2.11. Quantitative Real-Time Polymerase Chain Reaction (qRT-PCR)

RNAs were extracted from chondrocytes using the TaKaRa RNA extraction kit, and the PrimeScript^TM^ cDNA synthesis kit (TaKaRa, Japan) was used to convert the extracted RNAs to cDNA in accordance with the manufacturer’s instructions. Primers (Table 3) and the AccuPower^®^-2XGreenStar ^®^ qPCR Master Mix (Bioneer, Republic of Korea) were used to measure the amount of mRNA expressed. The following primer sequences were used:

### 2.12. Immunocytochemistry (ICC) Analysis

Primary cultured chondrocytes were seeded on coverslips (SPL, Pyeongtaek, Republic of Korea) and treated with HT042, DEX, and HT042+DEX after 24 h. The ICC was performed according to the guidelines of the Dako REAL^TM^ EnVision Kit (Agilent, USA), which will be described briefly. After 24 h, the coverslip was blocked with Dako REAL^TM^ peroxidase-blocking solution (Agilent, USA) for 5 min and washed with wash buffer. The primary antibody, Ihh (CST, Miami, FL, USA), was incubated overnight at 4 °C. The next day, after washing with wash buffer, it was reacted it with Dako REAL^TM^ EnVision^TM^/HRP (Agilent, USA) for 30 min at RT. Then, it was washed it twice with washing buffer. It was reacted with substrate working solution (Dako REAL^TM^ Substrate Buffer + Dako REAL^TM^ DAB + chromogen) and washed with DW. Lastly, it was mounted with Dako mounting medium (Agilent, USA).

### 2.13. Gene Slicing

Gene silencing of GHR was achieved by RNA interference using small interfering RNA (siRNA). Chondrocytes were transfected with DEPC, scrambled control siRNA, and siRNA directed against GHR using AccFect^TM^ transfection reagent (Bioneer Inc., Daejeon, Republic of Korea) according to the manufacturer’s instructions. One day before transfection, cells were seeded in 500 mL of growth medium without antibiotics such that they were 30–50% confluent at the time of transfection. The transfected cells were cultured in DMEM containing 10% FBS for 72 h after transfection (Anti-GHR rabbit antibody was from CST). To determine the knockdown efficiency, it was evaluated both the mRNA and protein expression of Ihh in transfected chondrocytes by qRT-PCR, respectively.

### 2.14. Statistics

Statistical analysis was performed using GraphPad Prism^®^ 9.0 (GraphPad Software, La Jolla, CA, USA) with one-way ANOVA and Dunnett’s post hoc tests. The measurements were indicated as the mean ± standard error of the mean. # *p* < 0.05 vs. NT, ## *p* < 0.01 vs. NT, ### *p* < 0.001 vs. NT in the one-way ANOVA and Dunnett’s test and * *p* < 0.05 vs. DEX, ** *p* < 0.01 vs. DEX, *** *p* < 0.001 vs. DEX in the Student’s *t*-test.

## 3. Results

### 3.1. High-Performance Liquid Chromatography (HPLC) Analysis

HPLC was used to measure the formononetin, eleutheroside E, and shanzhiside methyl ester amounts of HT042. HT042 composed 4.19 ± 0.03 mg/g of formononetin, 1.50 ± 0.05 mg/g of eleutheroside E, and 0.08 ± 0.00 mg/g of shanzhiside methyl ester (Figure 1A). Figure 1B–D show the HPLC chromatograms of the three marker compounds detected in the sample, respectively.

### 3.2. Evaluation of MTB Length Measurement

To determine whether HT042 could promote MTB length and recover the length growth inhibited by DEX, it was evaluated MTB length over 21 days (Figure 2A). After 21 days of length growth, the HT042 group showed a significant increase of 24.75% over NT from day 7, while DEX showed a significant decrease of 38.8% over NT from day 3. On the other hand, HT042+DEX showed a significant reduction in length growth compared with NT on days 0–7, but after day 14, the length growth decreased by 25. Moreover, 42% was compared with NT and recovered to a similar extent as NT with a 3.44% difference on day 21. In addition, DEX and HT042+DEX showed a significant difference in length growth between the two groups after day 7 (Figure 2B–E).

### 3.3. Effect on the Growth Plate Height Using Staining

To determine the effect of HT042 and DEX on growth plate length growth, it was measured the length of MTB growth plates by cresyl violet (CV) staining (Figure 3A) and toluidine blue (TB) staining (Figure 3B). HT042 increased the length of the hypertrophic zone (HZ) and proliferative zone (PZ) by 46.7% and 57% each compared with NT, while HT042 decreased the length of the HZ and PZ by 18.3% and 31.4% each compared with NT. HT042+DEX was decreased the length by 9% in the HZ and increased it by 27% in the PZ, indicating that HT042 promoted growth plate length growth while DEX inhibited growth. It was also found that HT042+DEX promoted length growth in the PZ with a slight decrease in the HZ. HT042+DEX showed a significant increase when compared with DEX (Figure 3C,D).

### 3.4. Effect of HT042 on the Ihh Expressions in Growth Plates

We investigated the Ihh expression in the growth plate by an immunohistochemistry analysis. The IHC results showed that there was a high level of Ihh expression in the PZ. The Ihh expression was higher without significance in the HT042 and HT042+DEX groups than in the NT group (Figure 4).

### 3.5. Effect of HT042 on Chondrocyte Apoptosis in MTB

To determine whether cell death due to cell proliferation and hypertrophy inhibited by DEX was promoted, HT042 was added to MTB cultured for 21 days using the TUNEL assay kit. HT042 showed almost no apoptosis in the PZ, while DEX caused apoptosis in a large portion of the cells. It can be concluded that DEX+HT042 prevented apoptosis by acting as an anti-apoptotic agent compared with DEX (Figure 5).

### 3.6. Effect of HT042 on Cell Proliferation in the Growth Plate

To determine the effects of HT042 and DEX on chondrocyte proliferation, chondrocytes were injected with BrdU. The cell proliferation ability of chondrocytes cultured for 3 days was checked by injecting BrdU, and compared with NT, HT042 increased by 35.7%, DEX decreased by 13.1%, and HT042+DEX increased by 6.5%. In addition, after treating chondrocytes with NT, HT042, DEX, and HT042+DEX, the number of chondrocytes was checked with DAPI, and it was found that the number of chondrocytes was significantly different in each group (Figure 6).

### 3.7. Effects of HT042 on mRNA Expression in the Chondrocyte

The analysis of GHR, IGF-1R, PTHrP, Ihh, caspase-3, BCl-X, BAX, and BCl-2 mRNA levels (Figure 7A–H) in chondrocytes identified that the HT042 treatment significantly increased in the GHR, IGF-1R, PTHrP, Ihh, BCl-X, and BCl-2 by about 1.48, 1.44, 1.61, 0.47 times and decreased in the caspase-3 and BAX 2.98 and 2.22 times when compared with the NT group. On the other hand, DEX treatment significantly decreased in GHR, IGF-1R, PTHrP, Ihh, BCl-X, and BCl-2 and increased in caspase-3 and BAX when compared with the NT group. HT042+DEX was not significant when compared with NT except for Ihh, but all were significant when compared with DEX. This identified that HT042+DEX restored the growth activity inhibited by DEX to promote proliferation and hypertrophy and increased anti-apoptotic activity by inhibiting apoptosis factors.

### 3.8. Effects of HT042 on Protein Expression in the Chondrocyte

The protein expression of the GHR, IGF-1R, PTHrP, Ihh, caspase-3, BCl-X, BAX, and BCl-2 in Figure 8. Caspase-3, BCl-X, BAX, and BCl-2 were apoptosis factors. As illustrated by the Western blot images, HT042 increased the expression of GHR, IGF-1R, PTHrP, Ihh, BCl-X, and BCl-2 and reduced the expression of caspase-3 and BAX. These results suggested that HT042 may increase the number of receptors present in the growth plate, thereby promoting growth through proliferative, hypertrophic, and anti-apoptotic effects and reversing DEX-induced growth inhibition.

### 3.9. Effects of HT042 and DEX on Ihh–DNA Binding Activity

We identified Ihh translocation with the stimulation of HT042 and DEX in order to investigate the connection between Ihh activation and chondrocyte function produced by HT042 and DEX. Our results showed that Ihh translocated into the nucleus following HT042 and DEX induction, as verified by microscopy (Figure 9).

### 3.10. Effects of HT042 on Anti-Apoptosis

We determined the effects of HT042, DEX, and DEX+HT042 on chondrocyte apoptosis by measuring the in situ cell death in light of the regulatory roles that HT042 and DEX had on apoptosis in other cell types. Chondrocytes indicated induced cell death (Figure 10) compared with NT. NT and DEX caused most of the cells to die, while HT042 did not cause most of the cells to die. DEX+HT042 did cause cell death, but it was significantly more anti-apoptotic than DEX. It was speculating that HT042 prevented apoptosis and directed the progression of DEX-induced apoptosis.

### 3.11. Effects on HT042 and DEX on GHR in Chondrocyte

We initially transfected rat GHR siRNA into chondrocytes derived from fetal rat MTB to determine whether HT042 interacts functionally with GHR in chondrocytes. When compared with the NT group, chondrocytes transfected with GHR siRNA showed lower expression of Ihh mRNA by about 3.6 times and protein (Figure 11A,B). It was assessed the effects of HT042, DEX, and HT042+DEX on the proliferation, differentiation, and apoptosis of the transfected growth plate chondrocytes in order to determine the results seen in the entire MTB. HT042 and HT042+DEX enhanced chondrocyte proliferation and differentiation by 5.26 and 2.2 times each in chondrocytes isolated from MTB treated with control siRNA, as measured by Ihh mRNA expression (Figure 11C). These stimulatory effects of HT042 and DEX were abolished by co-treating HT042 with DEX or transfecting it with GHR siRNA.

## 4. Discussion

According to the results, HT042 can promote growth plate length and recover DEX-induced growth inhibition through the MTB model, which identified that HT042 acts directly on the growth plate. It was also verified that HT042 increased the number of GHR/ IGF-1R on the surface of the growth plate through the mRNA and protein expression level, increased the number of receptors reduced by DEX, and increased the expression level of Ihh through GHR siRNA. The activity of the Ihh/ PTHrP pathway, one of the important mechanisms of growth promotion, was checked to enhance cell proliferation, and HT042 regulated caspase-3, BCl-X, BAX, and BCl-2 factors. These results were validated using both in vivo and in vitro models.

In the present study, it was having investigated the effect of HT042 on growth plate length growth and the ability of HT042 to restore growth that is inhibited by DEX using an in vitro organ culture model of fetal rat MTB. The MTB in vitro organ model is a well-established physiological model that has been extensively used to study the endochondral ossification process during longitudinal bone development and the effects of various types of factors on postnatal growth [24,29,30]. Because growth is involved in a number of complex endocrine factors, including the canonical GH-IGF-1 axis and local paracrine/autocrine [19,20,31,32,33] To determine the direct effects of DEX on the GHR/IGF-1R of the growth plate, it was aimed to determine the growth-promoting effects of HT042 and the recovery of DEX-induced growth retardation in the MTB model [25,34]. Chagin et al. isolated fetal MTB to determine the effects of DEX and growth factors and found that DEX inhibited MTB growth and IGF-1 stimulated growth. In this experiment, MTB length measurement showed that HT042 significantly increased the length compared with the NT group, and DEX significantly decreased the length compared with the NT group. In addition, when comparing DEX and HT042+DEX, DEX showed a decrease in length growth due to growth inhibition, while HT042+DEX showed a significant difference between the two groups due to the recovery of growth inhibition caused by HT042.The HT042+DEX group showed a significant decrease in length from 0 to 7 days compared with the NT group, but it recovered to a level similar to NT when measured at 21 days. Furthermore, histological analysis of the MTB, HT042 showed an increase in PZ and HZ length, while DEX showed a decrease in each PZ and HZ. In contrast, HT042+DEX restored the length to NT levels. This identified that HT042 can recover the growth inhibited by DEX when treated with DEX.

To determine the direct effect of DEX on the growth plate, it was determined the mRNA and protein expression levels of GHR in the growth plate, and the siRNA of GHR was produced to suggest the expression of Ihh. The reduction of GHR on the growth plate surface by DEX decreases the average number of receptors per cell because DEX binds to GHR, and overstimulation causes GHR to be internalized into the cell and then be degraded by degradation factors such as lysosomes [16,32,34,35]. It also downregulates GHR mRNA expression and the binding capacity and inhibits the increase in homologous induced GHR expression, resulting in increased insensitivity to GH [15,32,36]. Jux et al. found that high doses and prolonged use of DEX downregulated growth hormone receptor mRNA expression and binding capacity in growth plate chondrocytes and inhibited the induced increase in growth hormone receptor and IGF receptor expression. In this study, it was examined the mRNA and protein expression levels of GHR and found that HT042 was significantly increased compared with NT, while DEX was significantly decreased, and HT042+DEX was restored to a level similar to NT. It was also found that GHR siRNA inhibited the expression of Ihh, a downstream signaling involved in growth promotion. However, in the HT042 group, it was found that the Ihh mRNA expression level increased. This suggests that GHR restores resistance to DEX by restoring the internalization of GHR/IGF-1R and increasing receptor affinity and gene transcription.

In this study, to determine the effects of decreased GHR on the Ihh/PTHrP pathway, one of the important signaling pathways for growth promotion, it was analyzed the mRNA and protein expression levels of Ihh/PTHrP and examined their proliferation and differentiation. This is the first study to show that continuous DEX treatment attenuates longitudinal growth and growth plate chondrogenesis by inhibiting chondrocyte proliferation and terminal differentiation and activates growth plate chondrocyte apoptosis in the hypertrophic layer by inhibiting Ihh/PTHrP signaling, and HT042 restores these mechanisms. Ihh is a protein secreted by pre-hypertrophic chondrocytes in the growth plate during postnatal skeletal development and is a key regulator of endochondral ossification [8,37,38,39]. The Ihh/PTHrP pathway promoted a negative feedback loop after chondrocyte proliferation by Ihh, whereby Ihh signaling regulated intra-chondral ossification in conjunction with PTHrP signaling [40,41,42,43]. Peng et al. identified that glucocorticoid treatment during skeletal growth in newborn rats attenuated longitudinal growth and growth plate chondrogenesis by inhibiting chondrocyte proliferation and terminal differentiation, and it activated growth plate chondrocyte apoptosis in the hypertrophic layer by inhibiting Ihh/PTHrP signaling. In this experiment, it was measured the mRNA and protein expression levels after treatment with HT042, DEX, and HT042+DEX in chondrocyte cultures and found that HT042 significantly increased the expression of Ihh and PTHrP compared with NT, while DEX significantly decreased. In the case of HT042+DEX, Ihh and PTHrP were slightly decreased compared with NT in contrast to the DEX group, and the decreased levels of Ihh and PTHrP induced by DEX were restored. BrdU was measured to check proliferation, and HT042 showed a significant increase in BrdU compared with NT, while DEX showed a significant decrease. In the case of HT042+DEX, there was a slight increase compared with NT in contrast to the DEX group, indicating that the decreased proliferation caused by DEX was restored. To verify the recovery of Ihh/PTHrP pathway downregulation, the IHC analysis identified the retention of Ihh and PTHrP and showed that HT042 promoted Ihh/PTHrP expression in cultured rat metatarsal bone. The results showed that the growth retardation of longitudinal bone was observed after DEX treatment, suggesting that DEX-induced apoptotic chondrocytes in the hypertrophic zone interfered with the maturation and differentiation of chondrocytes into osteoblasts, resulting in weakened endochondral ossification and the downregulation of Ihh. Furthermore, HT042 treatment restored chondrocyte proliferation and differentiation, suggesting that HT042 affects chondrocyte proliferation and differentiation through the Ihh/PTHrP pathway.

In this study, we investigated whether HT042 inhibited proliferation and differentiation due to the expression of Ihh/PTHrP, which is inhibited by DEX, leading to chondrocyte apoptosis [44,45,46]. Peng et al. observed Ihh and PTHrP protein levels in the growth plate were decreased. Moreover, it was found that downregulating Ihh using siRNA increased chondrocyte apoptosis and inhibited PTHrP, SOX9, and Col2a1 expression. Chondrocytes were incubated with HT042, DEX, and HT042+DEX, and the mRNA expression were determined of Bax, Bcl-2, Bcl-X, and caspase-3 The mRNA expression of apoptosis-related factors Bax, Bcl-2, Bcl-X, and caspase-3 was significantly increased by HT042 compared with NT, and Bax was significantly decreased by Bcl-2, Bcl-X, and caspase-3. On the other hand, DEX significantly decreased Bax and significantly increased Bcl-2, Bcl-X, and caspase-3 compared with NT. In the case of HT042+DEX, Bax increased and Bcl-2, Bcl-X, and caspase-3 decreased compared with NT in contrast to the DEX group. In addition, the cultured cells were subjected to a TUNEL assay to determine which cells were apoptotic, and it was found that HT042 caused little apoptosis, while DEX induced apoptosis in most cells. In the HT042+DEX group, some apoptosis occurred, but a significant number of cells did not experience apoptosis. HT042 appears to enhance proliferation, differentiation, and anti-apoptosis by increasing the DEX-induced inhibition of the Ihh/PTHrP signaling pathway and inhibiting apoptosis.

## 5. Conclusions

HT042 promoted the length of growth plate and increased chondrocyte proliferation in metatarsal bone (MTB) model. HT042 also recovery the growth retardation by DEX after 21 days of HT042+DEX in the MTB model. HT042 restored GHR/IGF-1R, which was internalized and reduced by DEX, by increasing the number of receptors. It also restores proliferation and differentiation of inhibition by DEX as activating the Ihh/PTHrP signaling pathway. Histological analysis also identified that HT042 inhibited cell death by reducing apoptotic factors and promoting anti-apoptotic factors. This study suggests that HT042 could be a convenient, long-term treatment for children whose growth is retarded by long-term administration of DEX. 

## Figures and Tables

**Figure 1 nutrients-16-02333-f001:**
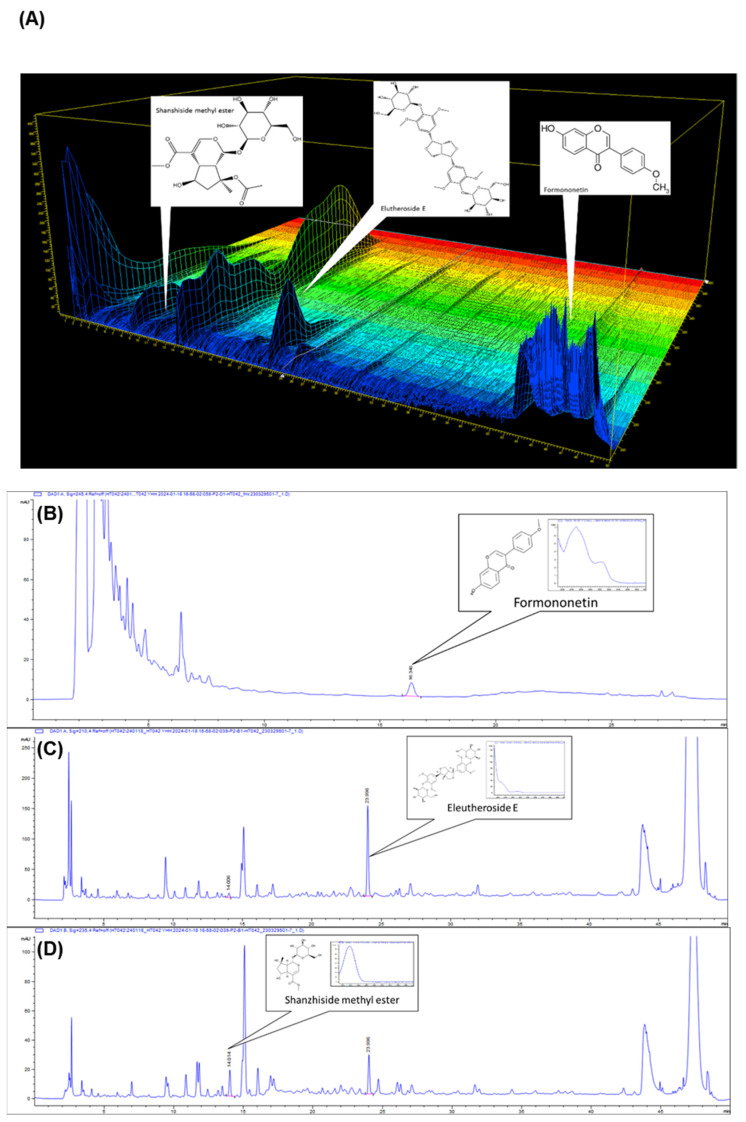
The 3D high-performance liquid chromatography chromatograms (HPLC) of formononetin, eleutheroside E, and shanzhiside methyl ester (**A**) in HT042. The HPLC shows formononetin (**B**), eleutheroside E (**C**) and shanzhiside methyl ester (**D**) as the compounds of *Astragalus membranaceus*, *Eleutherococcus senticosus*, and *Phlomis umbrosa* in HT042, respectively.

**Figure 2 nutrients-16-02333-f002:**
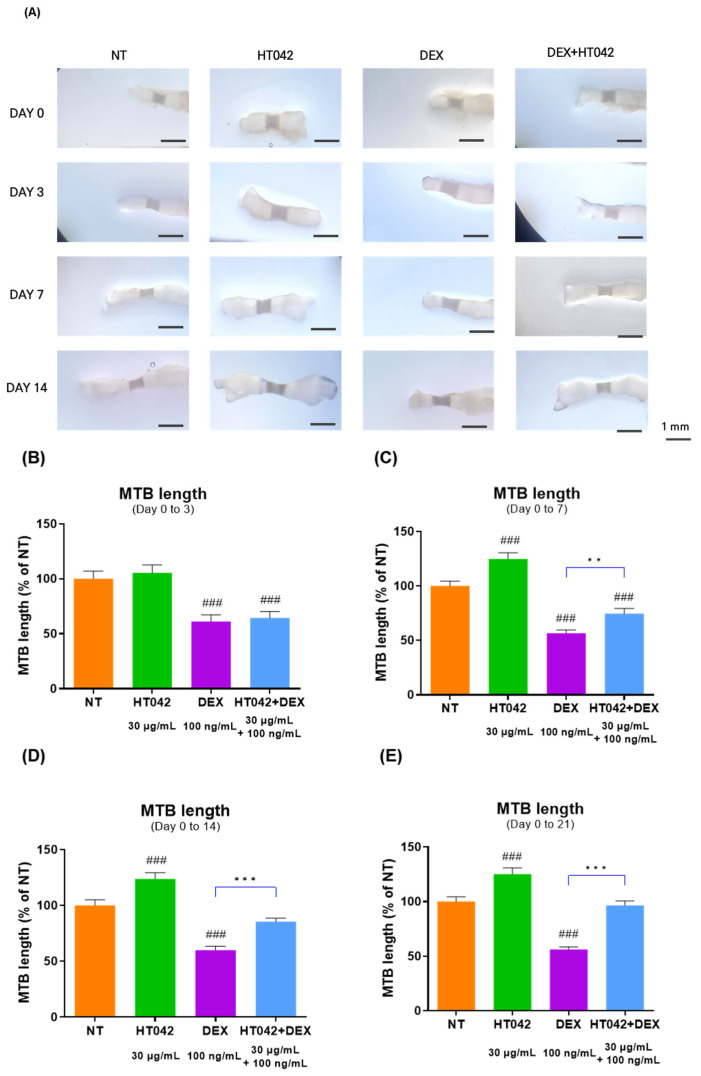
Evaluation of MTB length during 0 to 21 days. (**A**) Representative picture; (**B**–**E**) analysis of MTB length. Cells were treated with HT042 (30 µg/mL), DEX (100 ng/mL), and HT042+DEX (30 µg/mL + 100 ng/mL). ### *p* < 0.001 vs. NT by one-way ANOVA and Dunnett’s test; ** *p* < 0.01 vs. DEX; *** *p* < 0.001 vs. DEX by Student’s *t*-test. NT: Non-treated; MTB: metatarsal bone.

**Figure 3 nutrients-16-02333-f003:**
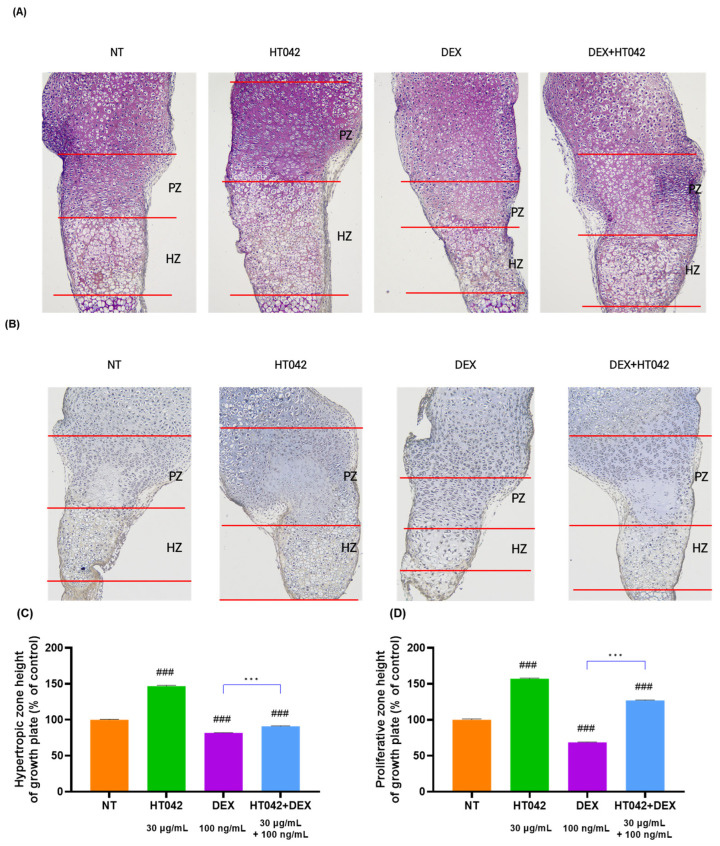
Photography of CV- and TB-stained growth plate of MTB. Representative photographs are shown from each treatment of MTB stained with (**A**) CV and (**B**) TB. Analysis of HZ and PZ length in each group (**C**,**D**). Cells were treated with HT042 (30 µg/mL), DEX (100 ng/mL), and HT042+DEX (30 µg/mL + 100 ng/mL). ### *p* < 0.001 vs. NT by one-way ANOVA and Dunnett’s test; and *** *p* < 0.001 vs. DEX by Student’s *t*-test. CV: Cresyl violet; HZ: hypertrophic zone; PZ: proliferative zone; TB: toluidine blue; NT: non-treated.

**Figure 4 nutrients-16-02333-f004:**
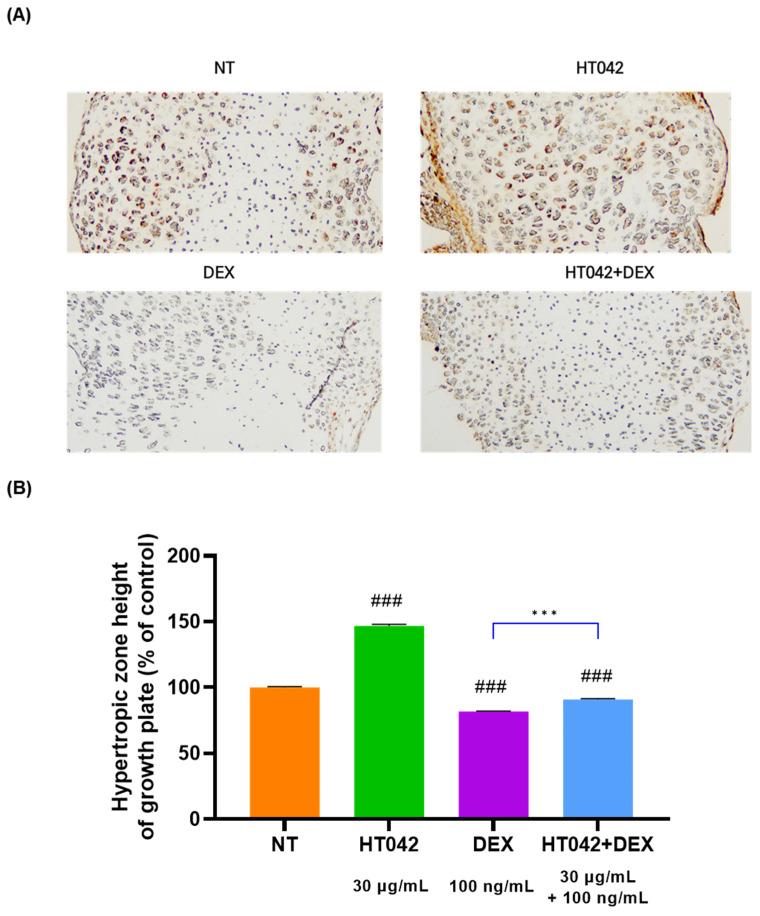
Immunohistochemical localization of Ihh in the growth plate. (**A**) representative photo of HZ of MTB. (**B**) analysis for HZ height of growth plate. Cells were treated with HT042 (30 µg/mL), DEX (100 ng/mL), and HT042+DEX (30 µg/mL + 100 ng/mL). ### *p* < 0.001 vs. NT by one-way ANOVA and Dunnett’s test; and *** *p* < 0.001 vs. DEX by Student’s *t*-test. IHC: immunohistochemistry; Ihh: Indian hedgehog; MTB: metatarsal bone, NT: non-treated; HZ: hypertrophic zone.

**Figure 5 nutrients-16-02333-f005:**
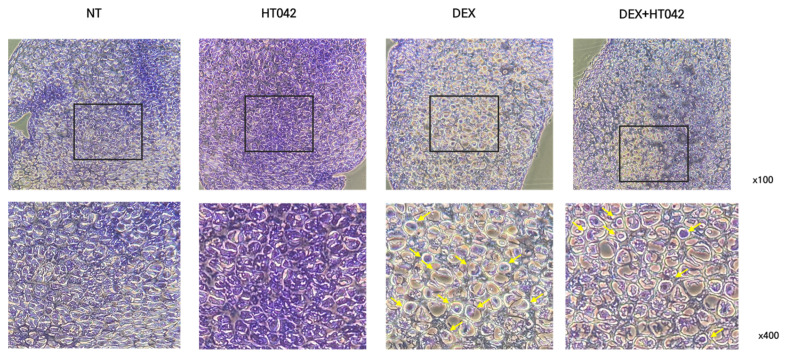
Effect of HT042 on anti-apoptosis in MTB. Apoptotic cells in the MTB were evaluated by the TUNEL assay. Representative photomicrograph indicated apoptosis in the MTB culture. Cells were treated with HT042 (30 µg/mL), DEX (100 ng/mL), and HT042+DEX (30 µg/mL + 100 ng/mL). MTB: Metatarsal bone, PZ: proliferative zone, TUNEL: terminal deoxynucleotidyltransferase-mediated dUTP end labeling.

**Figure 6 nutrients-16-02333-f006:**
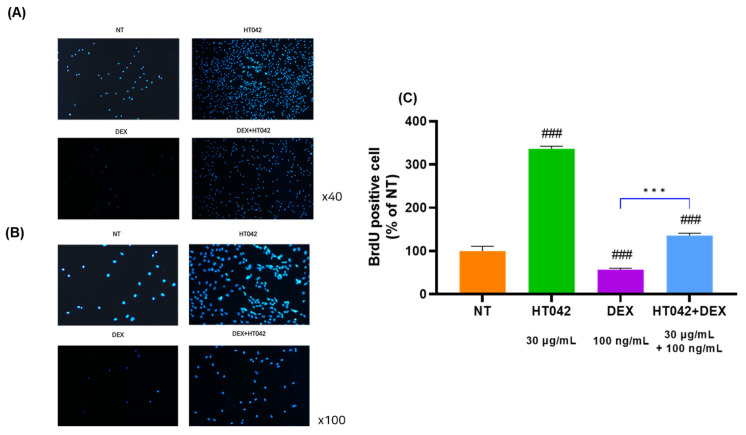
Effect of HT042 on cell proliferation in the MTB. (**A**) Chondrocytes were injected with BrdU in NT, HT042, DEX, and HT042+DEX. (**B**) Checking the cell proliferative effect using DAPI staining. (**C**) Analyzed for BrdU positive cell by % of NT. Cells were treated with HT042 (30 µg/mL), DEX (100 ng/mL), and HT042+DEX (30 µg/mL + 100 ng/mL). ### *p* < 0.001 vs. NT by one-way ANOVA and Dunnett’s test; *** *p* < 0.001 vs. DEX by Student’s *t*-test. BrdU: Bromodeoxyuridine; MTB: metatarsal bone; NT: non-treated.

**Figure 7 nutrients-16-02333-f007:**
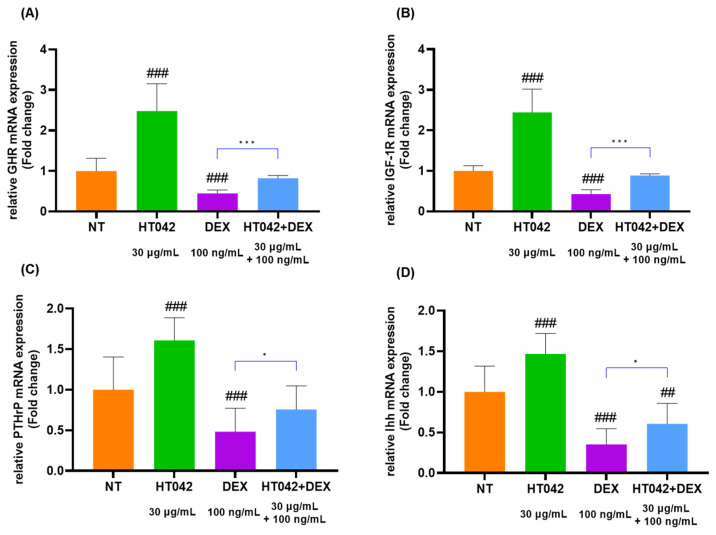
mRNA expression analysis in the chondrocyte. (**A**–**H**) mRNA expression of GHR, IGF-1R, PTHrP, Ihh, caspase-3, BCl-X, BAX, and BCl-2 as determined by qRT-PCR. Cells were treated with HT042 (30 µg/mL), DEX (100 ng/mL), and HT042+DEX (30 µg/mL + 100 ng/mL). ## *p* < 0.01 vs. NT, ### *p* < 0.001 vs. NT by one-way ANOVA and Dunnett’s test; * *p* < 0.05 vs. DEX and *** *p* < 0.001 vs. DEX by Student’s *t*-test. GHR: GH receptor; IGF-1R: insulin-like growth factor-1 receptor; NT: non-treated.

**Figure 8 nutrients-16-02333-f008:**
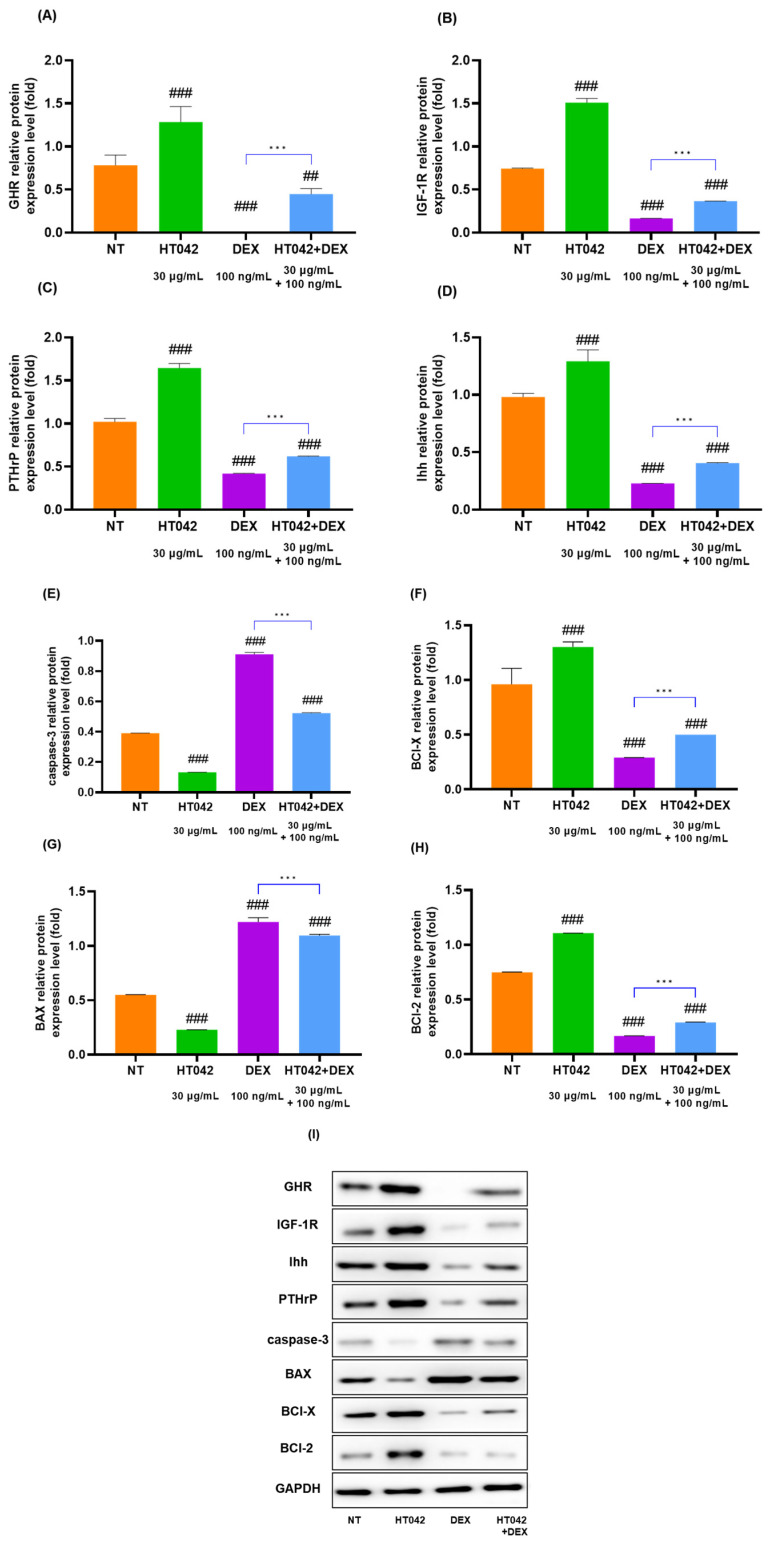
Protein expression of (**A**–**I**) GHR, IGF-1R, Ihh, PTHrP, caspase-3, BAX, BCl-X, BCl-2, and GAPDH in primary chondrocytes. Cells were treated with HT042 (30 µg/mL), DEX (100 ng/mL), and HT042+DEX (30 µg/mL + 100 ng/mL) for 72 h. ## *p* < 0.01 vs. NT, ### *p* < 0.001 vs. NT by one-way ANOVA and Dunnett’s test; *** *p* < 0.001 vs. DEX by Student’s *t*-test. GHR: GH receptor; IGF-1R: insulin-like growth factor-1 receptor; NT: non-treated.

**Figure 9 nutrients-16-02333-f009:**
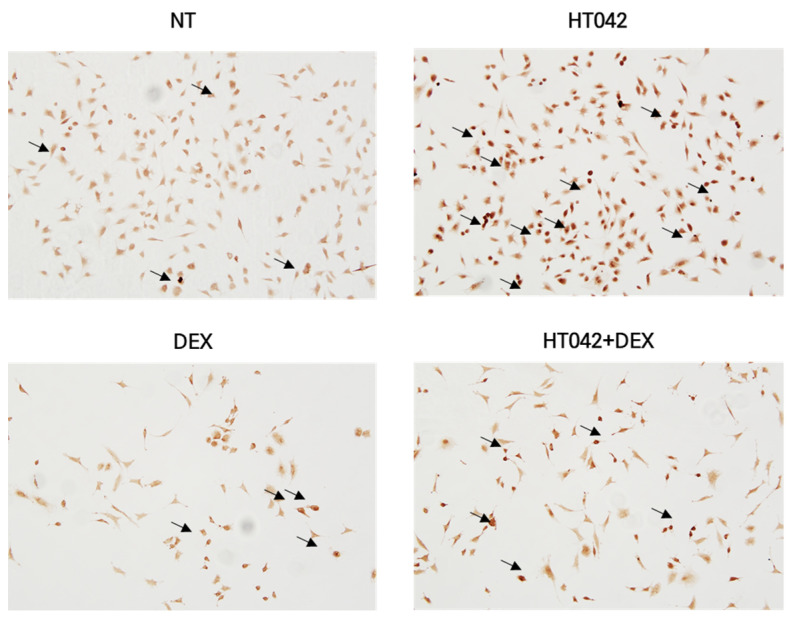
Effects of HT042 on Ihh/PTHrP signaling on Ihh target gene function. A representative interacted cell is indicated by the arrow. Cells were treated with HT042 (30 µg/mL), DEX (100 ng/mL), and HT042+DEX (30 µg/mL + 100 ng/mL). Ihh: Indian hedgehog; NT: non-treated.

**Figure 10 nutrients-16-02333-f010:**
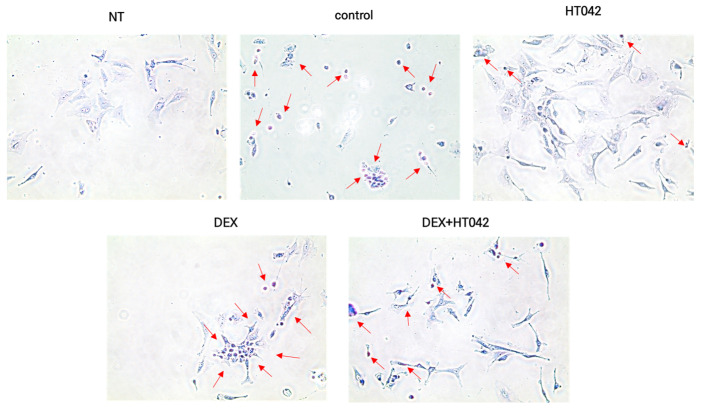
Effects of HT042 on chondrocyte apoptosis. A representative apoptotic cell is indicated by the arrow. Cells were treated with HT042 (30 µg/mL), DEX (100 ng/mL), and HT042+DEX (30 µg/mL + 100 ng/mL). NT: Non-treated; DEX: dexamethasone.

**Figure 11 nutrients-16-02333-f011:**
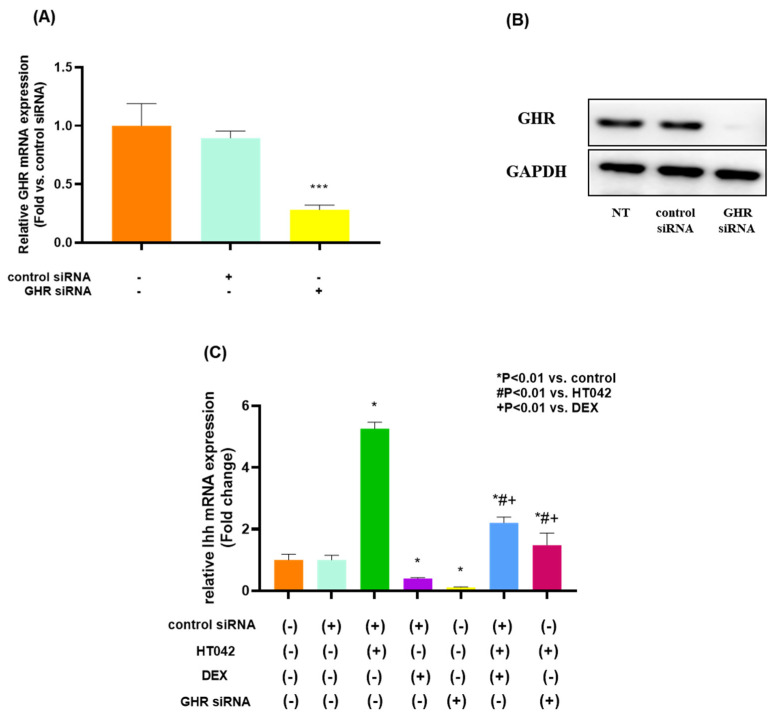
The effect of HT042 and DEX on chondrocyte proliferation and Ihh expression in cultured growth plate chondrocytes using transfected GHR siRNA. Chondrocytes were transfected with siRNA targeted for GHR and cultured in DMEM containing 10% FBS for 72 h after transfection. Both mRNA and protein expression were analyzed by real-time PCR (**A**) and Western blotting (**B**), respectively. *** *p* < 0.001 vs. NT by one-way ANOVA, Dunnett’s test. Cells were treated with HT042 (30 µg/mL), DEX (100 ng/mL), and HT042+DEX (30 µg/mL + 100 ng/mL). Ihh mRNA expression was detected in cultured chondrocytes by real-time PCR. Results are expressed as the as-fold change compared with untreated control chondrocytes (**C**). GHR: GH receptor; IGF-1R: insulin-like growth factor-1 receptor; NT: non-treated.

**Table 1 nutrients-16-02333-t001:** HPLC analysis condition for HT042.

	Condition
Column	SunFire C18 column (4.6 × 250 mm, 5 µm, Waters)
Mobile phase	(A) 0.5% phosphoric acid (B) acetonitrile
Flow rate	Eleutheroside E and shanzhiside methyl ester, 5% (0–20 min), 17% (20–30 min), 22% (30–40 min), 30% (40–45 min), 5% (45 min); Formononetin, 35% (0–15 min), 35% (15–25 min), 65% (25–28 min), 35% (28–30 min), 35% (30 min)
Injection volume	1.0 mL/min
Detection wavelength	Eleutheroside E: 235 nm Shanzhiside methyl ester: 210 nm Formononetin: 245 nm
Temperature	40 °C

**Table 2 nutrients-16-02333-t002:** HPLC analysis condition for formononetin, eleutheroside E, and shanzhiside methyl ester separately in HT042.

	Condition		
Compound	Formononetin	Eleutheroside E	Shanzhiside Methyl Ester
Column	SunFire C18 column (4.6 × 250 mm, 5 µm, Waters)	SunFire C18 column (4.6 × 250 mm, 5 µm, Waters)	SunFire C18 column (4.6 × 250 mm, 5 µm, Waters)
Mobile phase	(A) 0.5% phosphoric acid (B) acetonitrile	(A) 0.5% phosphoric acid (B) acetonitrile	(A) 0.5% phosphoric acid (B) acetonitrile
Flow rate	0–15 min (35%B), 15–25 min (35~65%B), 25–28 min (65~35%B), 28–30 (35%B)	0–20 min (5~17%B), 20–30 min (17~22%B), 30–40 min (22~30%B), 40–43 min (30~100%B), 43–45 min (100%B), 45–47 min (100~5%B), 47–50 min (5%B)	0–20 min (5~17%B), 20–30 min (17~22%B), 30–40 min (22~30%B), 40–43 min (30~100%B), 43–45 min (100%B), 45–47 min (100~5%B), 47–50 min (5%B)
Injection volume	1.0 mL/min	1.0 mL/min	1.0 mL/min
Detection wavelength	245 nm	235 nm	210 nm
Temperature	40 °C	40 °C	40 °C

**Table 3 nutrients-16-02333-t003:** mRNA primer sequence for the quantitative real-time RT-PCR analysis.

Gene Name		Primer Sequence (5′ → 3′)
Ihh	Forward	ATGTCTCCCGCCTGGCTC
	Reverse	TGGCGCCCAGGGTCTTCT
PTHrP	Forward	ATGCTGCGGAGGCTGGTT
	Reverse	GTCTTGGATGGACTTGCCC
Caspase-3	Forward	ATGGACAACAACGAAACCTC
	Reverse	CCAGATATATTCCAGAGTCC
Bcl2	Forward	AGTGGGATACTGGAGATGAA
	Reverse	TCAGGCTGGAAGGAGAAGAT
Bclx	Forward	CAGCTGGAGTCAGTTTAGCG
	Reverse	AAACTGCTGCTGTGGCCAGT
Bax	Forward	AGACACCTGAGCTGACCTTG
	Reverse	ATCAGCAATCATCCTCTGCA
GHR	Forward	ATGCTACAGACCAAGACACC
	Reverse	ACCCGCCAAAGATCCATA
IGF-1R	Forward	TTCTACAATTACGCACTGG
	Reverse	CTATGGTGGAGAGGTAACA
GAPDH	Forward	CTTGTGACAAAGTGGACATTGTT
	Reverse	TGACCAGCTTCCCATTCTC

Ihh: Indian hedgehog, PTHrP: parathyroid hormone-related protein, Bcl: B-cell lymphoma, Bax: Bcl-2 antagonist X, GHR: growth hormone receptor, IGF-1R: insulin-like growth factor-1 receptor, GAPDH: glyceraldehyde 3-phosphate dehydrogenase.

## Data Availability

All data is contained within the article.

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
