# Peer review of "Astragalus Extract Mixture HT042 Alleviates Dexamethasone-Induced Bone Growth Retardation in Rat Metatarsal Bones"

_nutrients, 2024, doi:10.3390/nu16142333_

Round 1
Reviewer 1 Report
Comments and Suggestions for Authors
This manuscript aimed to determine whether HT042 can counteract the ability of 99DEX to inhibit growth in a model in which metatarsal bone is treated with DEX to cause growth inhibition. The work was well conceived, and the data were logically analyzed. However, some errors in the manuscript should be revised by the author.
(1) There were no significant difference markers in some figures, such as Figure 2, 7, and 11. It is difficult for reviewers to judge whether there are significant differences within and between groups. The author needs to modify it.
(2) Line 277, Title 3.4 is incomplete, please to modify it.
(3) Line 29, there are 9 keywords in present manuscript, some keywords are not appropriate, such as growth plate, recovery, and GH receptor, etc. please revise the “keywords” section, and limit it to 5-6 keywords.
(4) Few literatures in this manuscript are within past five years, more recent literatures are recommended.
Author Response
[Reports on manuscript revision: Manuscript Number nutrients-3068773]
21 Jun 2024
We thank the editor and referees of the Nutrients for taking their time to review our article. We have made some corrections and clarifications in the manuscript after going over the referee’s comments. The changes are summarized below:
- Response to Referee 1
This manuscript aimed to determine whether HT042 can counteract the ability of 99DEX to inhibit growth in a model in which metatarsal bone is treated with DEX to cause growth inhibition. The work was well conceived, and the data were logically analyzed. However, some errors in the manuscript should be revised by the author.
Referee’s comment 1: There were no significant difference markers in some figures, such as Figure 2, 7, and 11. It is difficult for reviewers to judge whether there are significant differences within and between groups. The author needs to modify it.
Answer: Following your advice, we've included values in the manuscript. (page. 7-8, line 247-256/ page. 11-12, line 318-328/ page. 16, line 373-384)
Referee’s comment 2: Line 277, Title 3.4 is incomplete, please to modify it.
Answer: Thank you for your advice. We've checked and fixed it. (page. 10, line 282)
Referee’s comment 3: Line 29, there are 9 keywords in present manuscript, some keywords are not appropriate, such as growth plate, recovery, and GH receptor, etc. please revise the “keywords” section, and limit it to 5-6 keywords.
Answer: Thank you for your advice. I have limited the keywords to 5 as you suggested. (page. 1, line 32)
- Keywords: HT042, dexamethasone, growth retardation, proliferative, hypertrophic, apoptosis
Referee’s comment 4: Few literatures in this manuscript are within past five years, more recent literatures are recommended.
Answer: Thank you for your advice. As you suggested, we have added references within the last 5 years.
- Tyler, A.; Bryan, M.A.; Zhou, C.; Mangione-Smith, R.; Williams, D.; Johnson, D.P.; Kenyon, C.C.; Rasooly, I.; Neubauer, H.C.; Wilson, K.M. Variation in Dexamethasone Dosing and Use Outcomes for Inpatient Croup. Hospital Pediatrics 2022, 12, 22–29, doi:10.1542/hpeds.2021-005854.
- Zhao, Y.; Celvin, B.; Denis, M.C.; Karagianni, N.; Aulin, C.; Zaman, F.; Sävendahl, L. TNF Overexpression and Dexamethasone Treatment Impair Chondrogenesis and Bone Growth in an Additive Manner. Sci Rep 2022, 12, 18189, doi:10.1038/s41598-022-22734-8.
- Minnetti, M.; Caiulo, S.; Ferrigno, R.; Baldini‐Ferroli, B.; Bottaro, G.; Gianfrilli, D.; Sbardella, E.; De Martino, M.C.; Savage, M.O. Abnormal Linear Growth in Paediatric Adrenal Diseases: Pathogenesis, Prevalence and Management. Clinical Endocrinology 2020, 92, 98–108, doi:10.1111/cen.14131.
- Yin, M.; Wang, J.; Zhang, J.; Wang, W.; Lu, W.; Xu, F.; Ma, X.; Lyu, S.; Chen, L.; Zhang, L.; et al. Transcription Analyses of Differentially Expressed mRNAs, lncRNAs, circRNAs, and miRNAs in the Growth Plate of Rats with Glucocorticoid-Induced Growth Retardation. PeerJ 2023, 11, e14603, doi:10.7717/peerj.14603.
- Ma, J.; Siminoski, K.; Alos, N.; Halton, J.; Ho, J.; Cummings, E.A.; Shenouda, N.; Matzinger, M.A.; Lentle, B.; Jaremko, J.L.; et al. Impact of Vertebral Fractures and Glucocorticoid Exposure on Height Deficits in Children During Treatment of Leukemia. The Journal of Clinical Endocrinology & Metabolism 2019, 104, 213–222, doi:10.1210/jc.2018-01083.
- Madeo, S.F.; Zagaroli, L.; Vandelli, S.; Calcaterra, V.; Crinò, A.; De Sanctis, L.; Faienza, M.F.; Fintini, D.; Guazzarotti, L.; Licenziati, M.R.; et al. Endocrine Features of Prader-Willi Syndrome: A Narrative Review Focusing on Genotype-Phenotype Correlation. Front. Endocrinol. 2024, 15, 1382583, doi:10.3389/fendo.2024.1382583.
- Hua, J.; Huang, J.; Li, G.; Lin, S.; Cui, L. Glucocorticoid Induced Bone Disorders in Children: Research Progress in Treatment Mechanisms. Front. Endocrinol. 2023, 14, 1119427, doi:10.3389/fendo.2023.1119427.
- Lim, D.W.; Lee, C. The Effects of Natural Product-Derived Extracts for Longitudinal Bone Growth: An Overview of In Vivo Experiments. IJMS 2023, 24, 16608, doi:10.3390/ijms242316608.
- Lee, D.; Kim, B.-H.; Lee, S.-H.; Cho, W.-Y.; Kim, Y.-S.; Kim, H. Effects of Astragalus Extract Mixture HT042 on Circulating IGF-1 Level and Growth Hormone Axis in Rats. Children 2021, 8, 975, doi:10.3390/children8110975.
- Wang, F.; Yang, Z.; He, W.; Song, Q.; Wang, K.; Zhou, Y. Effects of Icariin on the Proliferation and Osteogenic Differentiation of Human Amniotic Mesenchymal Stem Cells. J Orthop Surg Res 2020, 15, 578, doi:10.1186/s13018-020-02076-9.
- Chun, J.M.; Lee, A.Y.; Moon, B.C.; Choi, G.; Kim, J.-S. Effects of Dipsacus Asperoides and Phlomis Umbrosa Extracts in a Rat Model of Osteoarthritis. Plants 2021, 10, 2030, doi:10.3390/plants10102030.
- Zang, H.; Yang, Q.; Li, J. Eleutheroside B Protects against Acute Kidney Injury by Activating IGF Pathway. Molecules 2019, 24, 3876, doi:10.3390/molecules24213876.
- Peng, G.; Sun, H.; Jiang, H.; Wang, Q.; Gan, L.; Tian, Y.; Sun, J.; Wen, D.; Deng, J. Exogenous Growth Hormone Functionally Alleviates Glucocorticoid-Induced Longitudinal Bone Growth Retardation in Male Rats by Activating the Ihh/PTHrP Signaling Pathway. Molecular and Cellular Endocrinology 2022, 545, 111571, doi:10.1016/j.mce.2022.111571.
- Qiu, J.; Fan, X.; Ding, H.; Zhao, M.; Xu, T.; Lei, J.; Ji, B.; Zhuang, Z.; Gao, Q. Antenatal Dexamethasone Retarded Fetal Long Bones Growth and Development by Down-Regulating of Insulin-like Growth Factor 1 Signaling in Fetal Rats. Hum Exp Toxicol 2022, 41, 096032712110728, doi:10.1177/09603271211072870.
- Ramesh, S.; Sävendahl, L.; Madhuri, V.; Zaman, F. Radial Shock Waves Prevent Growth Retardation Caused by the Clinically Used Drug Vismodegib in Ex Vivo Cultured Bones. Sci Rep 2020, 10, 13400, doi:10.1038/s41598-020-69904-0.
- Velentza, L.; Wickström, M.; Kogner, P.; Ohlsson, C.; Zaman, F.; Sävendahl, L. Humanin Treatment Protects against Venetoclax-Induced Bone Growth Retardation in Ex Vivo Cultured Rat Bones. Journal of the Endocrine Society 2024, bvae009, doi:10.1210/jendso/bvae009.
- Ma, R.; Liu, S.; Qiao, T.; Li, D.; Zhang, R.; Guo, X. Fluoride Inhibits Longitudinal Bone Growth by Acting Directly at the Growth Plate in Cultured Neonatal Rat Metatarsal Bones. Biol Trace Elem Res 2020, 197, 522–532, doi:10.1007/s12011-019-01997-9.
- Zhang, Z.; Zaman, F.; Nava, T.S.; Aeppli, T.R.J.; Gutierrez-Farewik, E.M.; Kulachenko, A.; Sävendahl, L. Micromechanical Loading Studies in Ex Vivo Cultured Embryonic Rat Bones Enabled by a Newly Developed Portable Loading Device. Ann Biomed Eng 2023, 51, 2229–2236, doi:10.1007/s10439-023-03258-2.
- Toni, L.; Plachy, L.; Dusatkova, P.; Amaratunga, S.A.; Elblova, L.; Sumnik, Z.; Kolouskova, S.; Snajderova, M.; Obermannova, B.; Pruhova, S.; et al. The Genetic Landscape of Children Born Small for Gestational Age with Persistent Short Stature. Horm Res Paediatr 2024, 97, 40–52, doi:10.1159/000530521.
- Sun, Q.; Huang, J.; Tian, J.; Lv, C.; Li, Y.; Yu, S.; Liu, J.; Zhang, J. Key Roles of Gli1 and Ihh Signaling in Craniofacial Development. Stem Cells and Development 2024, 33, 251–261, doi:10.1089/scd.2024.0036.
- Etschmaier, V.; Üçal, M.; Lohberger, B.; Weinberg, A.; Schäfer, U. Ex Vivo Organotypic Bone Slice Culture Reveals Preferential Chondrogenesis after Sustained Growth Plate Injury. Cells & Development 2024, 203927, doi:10.1016/j.cdev.2024.203927.
- Leung, A.O.W.; Poon, A.C.H.; Wang, X.; Feng, C.; Chen, P.; Zheng, Z.; To, M.K.; Chan, W.C.W.; Cheung, M.; Chan, D. Suppression of Apoptosis Impairs Phalangeal Joint Formation in the Pathogenesis of Brachydactyly Type A1. Nat Commun 2024, 15, 2229, doi:10.1038/s41467-024-45053-0.
- Zaman, F.; Zhao, Y.; Celvin, B.; Mehta, H.H.; Wan, J.; Chrysis, D.; Ohlsson, C.; Fadeel, B.; Cohen, P.; Sävendahl, L. Humanin Is a Novel Regulator of Hedgehog Signaling and Prevents Glucocorticoid‐induced Bone Growth Impairment. FASEB j. 2019, 33, 4962–4974, doi:10.1096/fj.201801741R.
We hope the revised manuscript will better meet the requirement of the Nutrients Journal for publication. We thank you once again for the constructive review by the referees.
Respectfully,
Chae Yun Baek, Ph.D

Reviewer 2 Report
Comments and Suggestions for Authors
The presented research describes the impact of a Korean Food and Drug Administration-approved astragalus extract on the evolution of a dexamethasone-induced bone growth retardation in rat metatarsal bones. The presented data is interesting however, it is hard to read due to numerous abbreviations (they should be explained first in the text, even if you have a designated abbreviation part).
The tables should also be checked and explained to all readers (see Table 1, what is the significance of F and R?).
Regarding the HPLC analysis the method of the elution gradient presentation is unclear it seems you have two separate gradients at the same time. Please rephrase for a better understanding.
You did not explain what form the HT042 was presented or given by the company (liquid, powder, etc, with or without excipients, etc.).
There is no reference to the concentration or dose of HT042 that was used for the experiments. How did you decide to use it and why?
The HPLC chromatogram although seems impressive in 3D format, is unclear in regards to proper separation and identification of the specified compounds.
The discussion section is another presentation of the results. You need to integrate your data within previous or similar research. You need to explain the significance of your results in comparison to literature data.
Comments on the Quality of English Language
Some phrases need to be reformulated for a clearer understanding. Also, the phrase Standardized HT042 composed 0.36% of eleutheroside E, 0.15% of shanzhiside methyl ester, and 0.008% of formononetin, has no meaning and needs to be rephrased taking into account the English language rules for topic of the phrase.
Author Response
[Reports on manuscript revision: Manuscript Number nutrients-3068773]
21 Jun 2024
We thank the editor and referees of the Nutrients for taking their time to review our article. We have made some corrections and clarifications in the manuscript after going over the referee’s comments. The changes are summarized below:
- Response to Referee 2
The presented research describes the impact of a Korean Food and Drug Administration-approved astragalus extract on the evolution of a dexamethasone-induced bone growth retardation in rat metatarsal bones.
Referee’s comment 1: The presented data is interesting however, it is hard to read due to numerous abbreviations (they should be explained first in the text, even if you have a designated abbreviation part).
Answer: Thank you for your advice. I have added an explanation to the abbreviations in the manuscript as per your advice.
Referee’s comment 2: The tables should also be checked and explained to all readers (see Table 1, what is the significance of F and R?).
Answer: Thank you for your advice. I have added the description as per your advice.
- F: forward/ R: reverse (page 6, line 207)
Referee’s comment 3: Regarding the HPLC analysis the method of the elution gradient presentation is unclear it seems you have two separate gradients at the same time. Please rephrase for a better understanding.
Answer: Thank you for your comment. The components of HT042 include eleutheroside E, shanzhiside methyl ester, and formononetin. Eleutheroside E and shanzhizide methyl ester are analyzed by separating the components with the same gradient conditions and analyzing the components at wavelengths of 235 nm and 210 nm, respectively, as specified in Table.1. The formonetin is separated with the gradient conditions specified in Table.2 and analyzed at a wavelength of 245 nm. (page 3, line 121, Table 1)
Referee’s comment 4: You did not explain what form the HT042 was presented or given by the company (liquid, powder, etc, with or without excipients, etc.).
Answer: Thank you for your question. We received HT042 powder from NeuMed to use in our experiments and have included this information in the manuscript. (page. 3, line 115)
Referee’s comment 5: There is no reference to the concentration or dose of HT042 that was used for the experiments. How did you decide to use it and why?
Answer: Thank you for your question. Prior to this study, we treated MTBs with HT042 at concentrations ranging from low to high, and chose 30 ug/ml as the concentration that showed no toxicity and a significant difference in growth promotion compared to the untreated group.
Referee’s comment 6: The HPLC chromatogram although seems impressive in 3D format, is unclear in regards to proper separation and identification of the specified compounds.
Answer: Thank you for your comment. Following your advice, we have analyzed each of the three components specified at the wavelengths added them to the manuscript. (page 3, line 122, Table 2/page 7, line 241, figure 1B-D)
Referee’s comment 7: The discussion section is another presentation of the results. You need to integrate your data within previous or similar research. You need to explain the significance of your results in comparison to literature data.
Answer: Thank you for your advice, we have followed your advice and added a discussion in the Discussion section comparing this study to previous studies.
We hope the revised manuscript will better meet the requirement of the Nutrients Journal for publication. We thank you once again for the constructive review by the referees.
Respectfully,
Chae Yun Baek, Ph.D
Reviewer 3 Report
Comments and Suggestions for Authors
The paper by Chae Yun Baek and colleagues present a very interesting and well conducted study reporting the effect of HT042 on bone metabolism. In details, the study provides evidences about the efficacy of the mixture in reversing the retard of bone growth by of glucocorticoids by means of ex-vivo an in vitro experiments.
Several minor points need to be addressed before publication:
- The use of the abbreviation GH should be carafully check troughout the manuscript,
Author Response
[Reports on manuscript revision: Manuscript Number nutrients-3068773]
21 Jun 2024
We thank the editor and referees of the Nutrients for taking their time to review our article. We have made some corrections and clarifications in the manuscript after going over the referee’s comments. The changes are summarized below:
- Response to Referee 3
The paper by Chae Yun Baek and colleagues present a very interesting and well conducted study reporting the effect of HT042 on bone metabolism. In details, the study provides evidences about the efficacy of the mixture in reversing the retard of bone growth by of glucocorticoids by means of ex-vivo an in vitro experiments. Several minor points need to be addressed before publication:
Referee’s comment 1: -The use of the abbreviation GH should be carafully check troughout the manuscript,
Answer: Thank you for your advice. I have added an explanation to the abbreviations section as per your advice.
We hope the revised manuscript will better meet the requirement of the Nutrients Journal for publication. We thank you once again for the constructive review by the referees.
Respectfully,
Chae Yun Baek, Ph.D
